# A multivariate analysis to propose linear models for the stature estimation in the Sabahan young adult population

Hasanur Bin Khazri[1], Sadia Choudhury Shimmi[1], M. Tanveer Hossain Parash [1,2]*

1 Department of Biomedical Sciences, Faculty of Medicine and Health Sciences, Universiti Malaysia Sabah, Kota Kinabalu, Malaysia, 2 Borneo Medical and Health Research Centre, Faculty of Medicine and Health Sciences, Universiti Malaysia Sabah, Kota Kinabalu, Malaysia

* parash_cmc@ums.edu.my

**Data Availability Statement:** All relevant data are within the paper and its Supporting Information files.

## Abstract

### Background

Stature is one of the significant parameters to confirm a biological profile besides sex, age, and ancestry. Sabah is in the Eastern part of Malaysia and is populated by multi-ethnic groups. To date, limited studies on stature estimation have been conducted in Sabah. Hence, this study aims to construct population-specific stature estimation equations for the large ethnic groups in Sabah, Malaysia.

### Objective

The aim is to propose linear models using different hand dimensions (hand span, hand-breadth, hand length, middle finger length, and the second inter-crease in the middle finger) for the young adult male and females of the major ethnic groups in Sabah.

### Materials & methods

This cross-sectional study framework used stratified random sampling on 184 male and 184 female young adults. An unpaired t-test and a one-way ANOVA were used to assess the differences in the mean between sex and ethnicities, respectively. The link between the response variable and explanatory variables was initially investigated using simple linear regression, followed by multiple linear regression.

### Result

The present study demonstrated the highest association for the quantitative explanatory variables among hand length and stature (right side: r = 0.833; left side: r = 0.842). Simple equations were specifically developed without sex indicators, and ethnic and multiple linear regression was developed with sex and ethnic indicators. Multiple linear regression provided good estimation $r^2$ = 0.7886 and adjusted $r^2$ = 0.7853. The stature of 18 to 25 year old large ethnic groups in Sabah can be estimated using the developed models 90.218 + 3.845 LHL -5.950 Sex—2.308 Bajau -1.673 KadazanDusun + 2.676 L2ICL. While, formula for each

**Funding:** SCS and MTHP received grant.The Centre for Research and Innovation (PPI), Universiti Malaysia Sabah, funded this research work under the grant "Skim Geran Acculturation" (SGA0041-2019). The funders had no role in study design, data collection and analysis, decision to publish, or preparation of the manuscript.

**Competing interests:** The authors have declared that no competing interests exist.

ethnic and sex KadazanDusun Male: Stature = 88.545 + 3.845 LHL+ 2.676 L2ICL, KadazanDusun Female: Stature = 82.595 + 3.845 LHL+ 2.676 L2ICL, Bajau Male: Stature = 87.910 + 3.845 LHL+ 2.676 L2ICL, Bajau Female: Stature = 81.960 + 3.845 LHL+ 2.676 L2ICL, Malay Male: Stature = 90.218 + 3.845 LHL+ 2.676 L2ICL, Malay Female: Stature = 84.268 + 3.845 LHL+ 2.676 L2ICL, Chinese Male: Stature = 90.218 + 3.845 LHL+ 2.676 L2ICL, and Chinese Female: Stature = 84.268 + 3.845 LHL+ 2.676 L2ICL.

## Conclusion

The study reports anthropometric data and formulas for measuring the stature of major ethnic groups in Sabah, which can be used to compare future work.

## Introduction

Stature is one of the significant parameters to confirm a biological profile besides sex, age, and ancestry [1]. Stature is vital for nutrition and health in the calculation of body mass index (BMI) [2]. In numerous low- and middle-income countries, the substantial prevalence of short adult stature represents the nutrition net impact through time and across generations and the involvement of diseases and other related environmental variables, such as socioeconomic position [3]. Height prediction is essential in spine and limb deformities, surgical procedures or trauma, skeletal dysplasia, and measuring stature age-related loss [4]. Stature is considered one of the unique critical parameters for personal identification essential in forensic medicine cases [5].

Many studies using mathematical equations demonstrated that stature is estimated from other body parts [6, 7]. In an ongoing investigation, stature estimation can aid in narrowing down the potential victims in scenarios of person identification from the remains in forensic investigation processes, thus providing valuable clues for the investigating authorities to identify the suspects accurately [8].

Different studies adopted different approaches to estimate stature, such as from the lower limbs [9, 10], upper limbs [11–13], and head [14, 15]. Although foot length is very reliable for stature estimation [16], if the body parts are damaged or lost, and there are no feet attached, the stature estimation from the hand will be helpful. Studies on hand measurement are reliable in estimating stature [1, 11], and the right-hand length is the most reliable among these hand measurements [1].

Previous work developed a mathematical equation from regression analysis to determine stature from the hand [1, 13] and the hand and foot [16, 17]. Multiple regression analysis is better to find the stature from the foot breadth and length, as standard error of estimate (SEE) and coefficient values of determination were better than those in a simple linear regression equation [16]. Similarly, the multiple regression equation was preferred, as it had low values of SEE for upper limb stature estimation [1].

Ethnic variations in the population, nutrition, genetics, sex, environment, age, and physical activity, impact the stature [3, 7, 18–21]. Therefore, the formula designed for a particular population might not fit for others. The different populations studied the estimation of the stature from other regions such as Australia [22], Korea [16], Bangladesh [13], and East Malaysia [11]. Sabah is in the Eastern part of Malaysia and is populated by multi-ethnic groups. Sabah has over fifty main ethnic groups with their own languages. Among these ethnicities KadazanDusun speaking ethnicities are the largest followed by Bajau, Malay and other ethnic groups.

Among the non-indigenous groups, the Chinese are majority. The origin of the Malay and Chinese populations of this study is different from the West Malaysia. The Sabahan Malays are mostly of Bruneian and Kadayan origin [23] while in peninsular Malaysia, Malay subethnic groups are Melayu Kelantan, Melayu Minang, Melayu Jawa, and Melayu Bugis [24]. The Hakkas are the majority in Sabah among the Chinese population on top of few Cantonese, Hokkien, Teochew, Hainanese, and Shantung [23], whereas, in West Malaysia, Hokkien, Cantonese, Foochows, and other groups are prominent [25]. Until now, limited studies on stature estimation addressing the ethnic variation have been conducted in Sabah. Hence, this study aims to propose population-specific stature estimation equations for major ethnic groups in Sabah, Malaysia.

## Materials and methods

This study adopted a cross-sectional approach and was performed from February 2021 to January 2022. The study was designed to propose linear models using different hand dimensions (hand span, breadth, and length, as well as the middle finger and second inter-crease of the middle finger length) for the young adult male and female population of the large ethnic groups in Sabah.

### Study population

This study comprised four ethnic groups: Malay, Chinese, KadazanDusun, and Bajau. To fulfill the inclusion criteria, subjects' parents and grandparents must be from the same ethnic groups. The KadazanDusun and Malay ethnic groups are most abundant in Tuaran, Tamparuli, Ranau, and Papar districts, whereas the Bajau in Kota Belud, Tuaran, and Semporna, and the Chinese in the Kudat district. The standard operating procedure for preventing the Covid-19 pandemic spread did not permit collecting data at the community level. Alternatively, the study was carried out among university students from the districts mentioned above who completed their Covid-19 vaccination.

**Inclusion criteria.** The qualified participants considered for this study satisfied the following key criteria:

a. The minimum age was 18, and the maximum was 25 years.

b. They stay at the university campus.

c. They are Malay (Bruneian), Chinese Bajau, or KadazanDusun.

**Exclusion criteria.**

a. Individuals with Parkinsonism, rheumatoid arthritis, or other medical conditions that can impact hand anthropometry.

b. Individuals having parents or grandparents who were not from the same ethnic group.

### Data collection

The respondents obtained informed consent after explaining the study's design, objectives, and methodology. The researcher used a stadiometer to measure the heights of the respondents and an INSIZE (0-200mm x 0.01mm 1108–200) digital caliper to measure the hand dimensions. The values were reported in centimeters. The measurement was taken two times,

and the overall average was reported. The measurement was taken in the fixed period of 10 am to 12 pm to avoid the possible diurnal variation.

## Sample size

The minimum sample size recommended by the researcher was 25 participants for each stratum [26]. Predicting a response rate of about 50% from prior research in acquiring participants from the same target population, the researchers added 50 individuals for each stratum. A total of 368 participants acted as the target population in this study, amounting to 46 participants for each stratum [46x4 (ethnicity) x2 (sex)].

## Sampling of the subjects

Student Affairs Department (BPA) provided a list of 18 to 25-year-old students of Malay, Chinese, KadazanDusun, and Bajau ethnic groups. The students were segregated according to four ethnic groups and were further divided into the male and female categories. Fifty subjects were chosen randomly from every stratum until the required sample size was met.

## Stature measurement

Standing height was measured using a stationary stadiometer with an upright backboard and a movable headboard. After removing the footwear, subjects stood together on the stadiometer's platform with their heels. The arms hung at the sides, with the palms facing the thighs and the gaze directed straight forward. The back was as straight as possible, so the heels, glutes, shoulders, and head all contacted the instrument's vertical section. The subject's head was aligned with the horizontal Frankfort plane. Along the midsagittal plane, the head plates of the stadiometer were put into solid contact with the vertex. After instructing the participant to breathe deeply and retain their breath, readings to the closest 0.1 cm [27] were collected.

## Measurement of hand dimensions

**Hand length.**   The distance from the midpoint of the distal wrist crease to the most distal point of the middle finger was measured as the hand's length [28].

**Handbreadth.**   The distance from the lateral surface of the second metacarpal and the medial surface of the fifth at the knuckles was measured as the handbreadth. [21].

**Middle finger length.**   The middle finger length was considered to be from the proximal finger crease of the middle finger to its tip [29].

**Second inter-crease length of the middle finger.**   The middle phalanx was determined as the distance from the distal interphalangeal joint crease to the proximal interphalangeal joint crease [30].

**Handspan.**   The linear distance between the thumb's tip and the small finger's tip in the hand spread wide was considered the handspan [31].

All the dimensions were measured in cm.

## Statistical analysis

The mean scores were utilized to define the height and hand dimensions as the data had a normal distribution. An unpaired t-test and a one-way ANOVA were used to assess the differences in the mean between sexes and ethnicities, respectively. The link between the response variable and explanatory variables was initially investigated using simple linear regression, followed by multiple linear regression. Pearson's correlation test investigated the multicollinearity of the variables. R-squared, adjusted R-squared, and estimated standard errors were used to

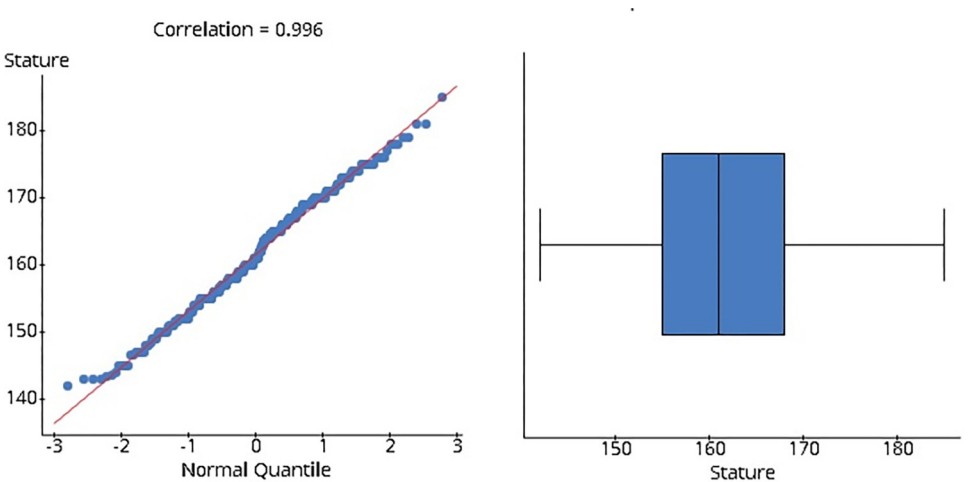

**Fig 1. Q-Q plot with correlation with normal quantile and boxplot of stature.**

assess the goodness-of-fit of various models. The statistical analysis software applications employed were IBM SPSS Statistics 28 and StatCrunch. The significance threshold used was α = 0.05.

### Ethics statement

The study received ethical approval from the Medical Research Ethics Committee. UMS. The committee's reference number is JKEtika 5/20(7). Written informed consent was obtained from all the participants before data collection.

## Results

The measured statures of the participants are normally distributed (Fig 1). The mean and standard deviation of the participants' statures were distributed among the Malay, Chinese, KadazanDusun, and Bajau populations in Sabah, which were further subdivided by sex (Table 1).

From the distribution among the sexes in each ethnic group, male participants had higher stature than female participants (Table 1). An independent sample t-test was performed to investigate the statistical significance of the observed difference. The participants' inclusion in the study was random, and independent observations were made. Each group contained over 30 participants, which is not over 5% of the target population. Hence, the assumptions for the t-test were fulfilled [32].

**Table 1. Stature of the respondents based on sex and ethnicity (n = 368).**

| Ethnicity | Sex | Stature | |
|---|---|---|---|
| | | Mean | SD |
| KadazanDusun | Male | 167.24 | 5.18 |
| | Female | 153.45 | 5.32 |
| Bajau | Male | 163.72 | 6.16 |
| | Female | 153.70 | 6.17 |
| Malay | Male | 169.48 | 5.58 |
| | Female | 156.15 | 5.77 |
| Chinese | Male | 169.22 | 4.58 |
| | Female | 158.66 | 5.98 |

**Table 2. Difference between stature and hand dimensions among sex (n = 368).**

| | Mean Difference | Standard error | t | df | p-value | 95% CI of the Difference | |
|---|---|---|---|---|---|---|---|
| | | | | | | Lower | Upper |
| **Stature** | 11.92 | 0.62 | 19.096 | 367 | <0.001 | 10.70 | 13.15 |

The one-way AONVA test assessed the apparent difference in mean scores of statures for the ethnicities (Table 1) to test the following hypothesis:

Based on this, the following hypothesis was formulated:

$H_0$: There is no difference in stature between the male and female participants.

$H_1$: Male participants' stature was higher than female participants.

The null hypothesis for the test was rejected, as the p-value (<0.001) for the t-statistics in terms of the difference of mean values for stature was less than the significance level, α = 0.05 (Table 2). Therefore, the result demonstrates that male participants had a higher stature than females.

$H_0$: There is no difference in stature among the Malay, Chinese, Bajau, and KadazanDusun participants.

$H_1$: At least one ethnic group of participants has different stature than others.

The normal probability plots with the correlation between the score and the Z-score for every ethnic group are illustrated in Fig 2. The correlations are greater than the critical value (0.960), so each data set stems from a normally distributed population [33].

The one-way ANOVA test has another assumption that each stratum must have a similar variance [32]. The standard deviations for each group were compared. The highest standard deviation, 6.17, was less than twice the lowest, 4.58 (4.58x2 = 9.16 > 6.17). So, the need for equal population variance values was achieved [32].

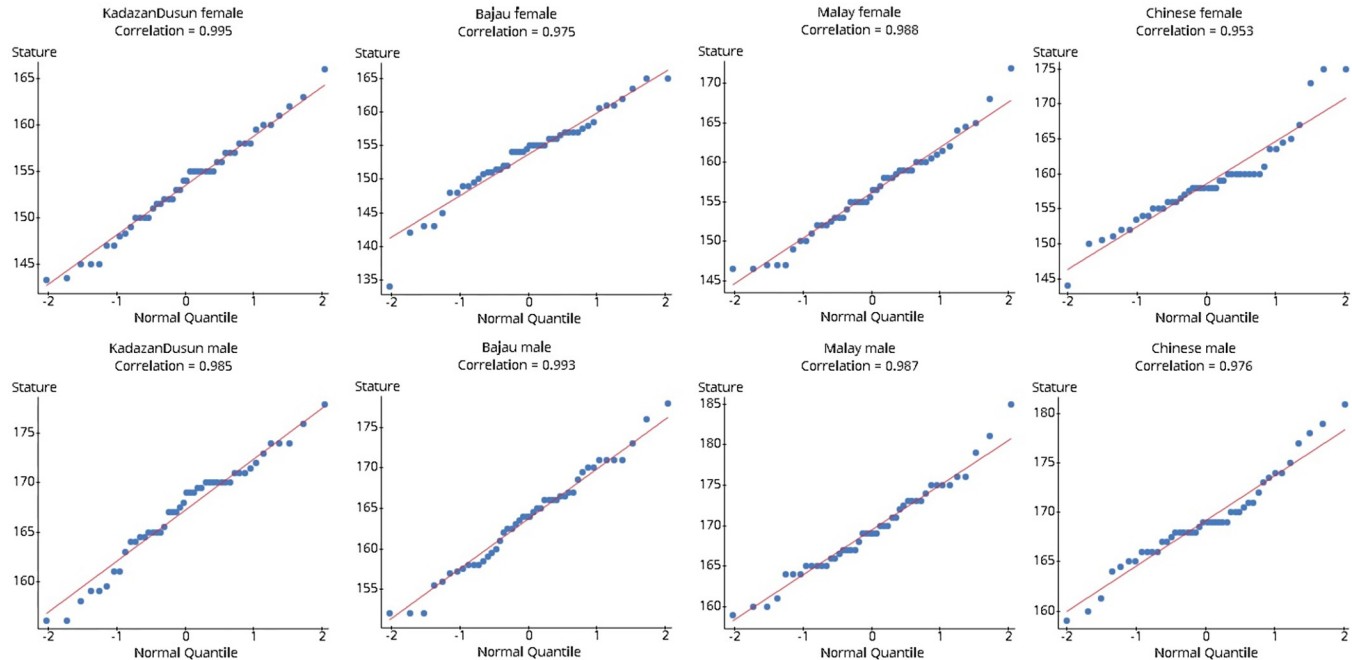

**Fig 2. Q-Q plot for correlation of observations of various ethnicities with z-scores.**

**Table 3. ANOVA results.**

| | | Sum of Squares | df | Mean Square value | F | Sig. |
|---|---|---|---|---|---|---|
| Male | Among Groups | 974.664 | 3 | 324.888 | 11.170 | < .001 |
| | Within Groups | 5235.655 | 180 | 29.087 | | |
| | Total | 6210.319 | 183 | | | |
| Female | Between Groups | 822.180 | 3 | 274.060 | 8.091 | < .001 |
| | Within Groups | 6097.327 | 180 | 33.874 | | |
| | Total | 6919.506 | 183 | | | |

Table 3 demonstrates that p-values (<0.001) for the difference in male and female statures of major ethnic groups are lower than the significance level ($\alpha = 0.05$). Therefore, the null hypothesis was rejected. There is enough proof to point out that the stature of at least one of the groups differs from the others. Therefore, the post hoc multiple comparisons test was required to investigate in which pair the difference exists. Tukey's post hoc honestly significant difference (i.e., HSD) assessment was conducted, as the data satisfied the assumption of the homogeneity of the variance values.

The p-values in Table 4 indicate a significant difference in the stature of Bajau males and KadazanDusun ($p<0.05$), Malay ($p<0.001$), and Chinese ($p<0.001$) males, where Bajau males' average stature was the lower than all these ethnicities. Again, Chinese females were significantly ($p<0.001$) taller than KadazanDusun and Bajau females.

At first, the relation between hand dimensions and stature was evaluated by a linear regression model, and after that, a multiple linear regression. The assumption of random selection for linear regression was achieved, as the subjects were selected through a stratified random sampling approach. Another assumption for the linear regression model is that there should be a linear relationship between the response variable (i.e., stature) and the explanatory variable (i.e., individual hand dimensions). The residual distribution of every parameter was derived to determine the linear relation. The correlation values were also computed, as shown in Fig 3. All the residual values showed a reasonably linear relation with the Z-score. The correlation values among the residuals and the hand dimensions were all over the critical value (0.960) [33]. Therefore, we can say that, for every hand dimension, the stature was normally distributed.

**Table 4. Difference of stature in different ethnicities among the respondents (n = 368).**

| | I -Ethnicity | J—Ethnicity | Mean Difference score (I-J) | Standard Error | p-value | 95% Confidence interval | |
|---|---|---|---|---|---|---|---|
| | | | | | | Lower | Upper |
| Male | KadazanDusun | Bajau | 3.52 | 1.12 | 0.011 | 0.60 | 6.43 |
| | | Malay | -2.24 | 1.12 | 0.195 | -5.16 | 0.68 |
| | | Chinese | -1.98 | 1.12 | 0.294 | -4.90 | 0.93 |
| | Bajau | Malay | -5.76 | 1.12 | <0.001 | -8.67 | -2.84 |
| | | Chinese | -5.50 | 1.12 | <0.001 | -8.42 | -2.58 |
| | Malay | Chinese | 0.26 | 1.12 | 0.996 | -2.66 | 3.17 |
| Female | KadazanDusun | Bajau | -0.25 | 1.21 | 0.997 | -3.40 | 2.89 |
| | | Malay | -2.70 | 1.21 | 0.120 | -5.85 | 0.44 |
| | | Chinese | -5.22 | 1.21 | <0.001 | -8.36 | -2.07 |
| | Bajau | Malay | -2.45 | 1.21 | 0.185 | -5.60 | 0.70 |
| | | Chinese | -4.96 | 1.21 | <0.001 | -8.11 | -1.81 |
| | Malay | Chinese | -2.51 | 1.21 | 0.167 | -5.66 | 0.64 |

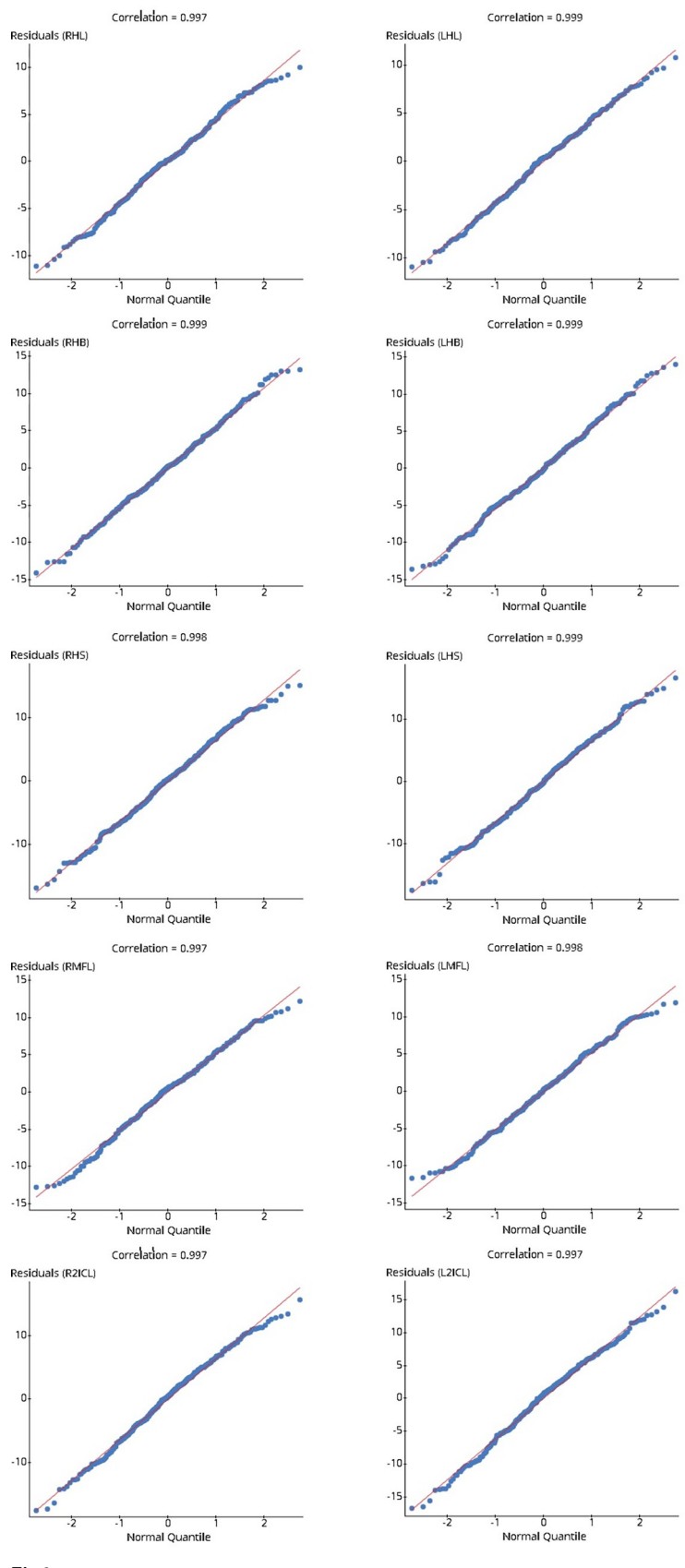

**Fig 3.**

Table 5. Constants, coefficients, and relationships of upper limb dimensions with stature ($n = 463$).

| Variable | Intercept ($\beta_0$) | Slope ($\beta_i$) | Correlation Coefficient (r) |
|---|---|---|---|
| RHL | 50.402 | 6.260 | 0.833* |
| LHL | 49.263 | 6.327 | 0.842* |
| RHB | 83.124 | 9.894 | 0.727* |
| LHB | 84.494 | 9.786 | 0.713* |
| RHS | 114.765 | 9.894 | 0.568* |
| LHS | 115.298 | 2.794 | 0.552* |
| RMFL | 67.758 | 12.267 | 0.752* |
| LMFL | 71.124 | 11.829 | 0.753* |
| R2ICL | 105.840 | 21.252 | 0.570* |
| L2ICL | 102.346 | 22.503 | 0.610* |

RHL = Right hand length, LHL = Left hand length, RHB = Right handbreadth, LHB = Left handbreadth,

RMFL = Right middle finger length, LMFL = Left middle finger length, R2ICL = Right second inter-crease length,

L2ICL = Left second inter-crease length, RHS = Right hand span, LHS = Left hand span

*Relationship is significant at p<0.0001 level.

The null hypothesis was rejected, as the p-value (<0.001) for the slope was lower than the significance level α = 0.05 (Table 5). There is sufficient evidence proof to point out that a linear relation exists for hand span, length, breadth, middle finger length, second inter-crease middle finger length, and stature.

The F-statistics values are under p<0.05, and according to Fig 4, no discernible pattern in the residual plots and no outlier in the boxplot (Fig 5) indicate the suitability of the model (Table 6).

Multiple linear regression requires normally distributed residuals and avoiding multicollinearity between the explanatory variables. The test also requires no outliers in the residuals to draw inferences from the multiple regression findings [33]. A correlation test examined the collinearity among the explanatory variables, as shown in the correlogram (Table 7).

The LHL had a high correlation with RHL and LMFL, whereas LHL had a higher correlation with stature than RHL and LMFL (Fig 6). RHL and LMFL were removed from the model to avoid multicollinearity. The multiple regression tested the following hypothesis.

$H_0$: There is no relation between LHL, LHB, LHS, L2ICL, RHB, RMFL, R2ICL, RHS, sex, ethnicity, and stature ($\beta_n = 0$).

$H_1$: A linear relation exists among hand dimensions, ethnicity, sex and stature (At least one $\beta_i \neq 0$).

The normality of the residuals assumptions, equal variance distribution, and unavailability of any outliers act as a pre-requisite for concluding the findings (Fig 6). Also, the null hypothesis was rejected, as the slope p-values under the significance level (Tables 8 and 9).

The F-statistics values are under p<0.05, and according to Fig 4, no discernible pattern in any of the residual plots and no outlier in the boxplot indicate the suitability of the model (Table 10).

If we place the code values for sex (male = 0; female = 1), Bajau (Bajau = 1, Others = 0) and KadazanDusun (KadazanDusun = 1, Others = 0) we obtain the following:

KadazanDusun Male: Stature = 88.545 + 3.845 LHL+ 2.676 L2ICL

KadazanDusun Female: Stature = 82.595 + 3.845 LHL+ 2.676 L2ICL

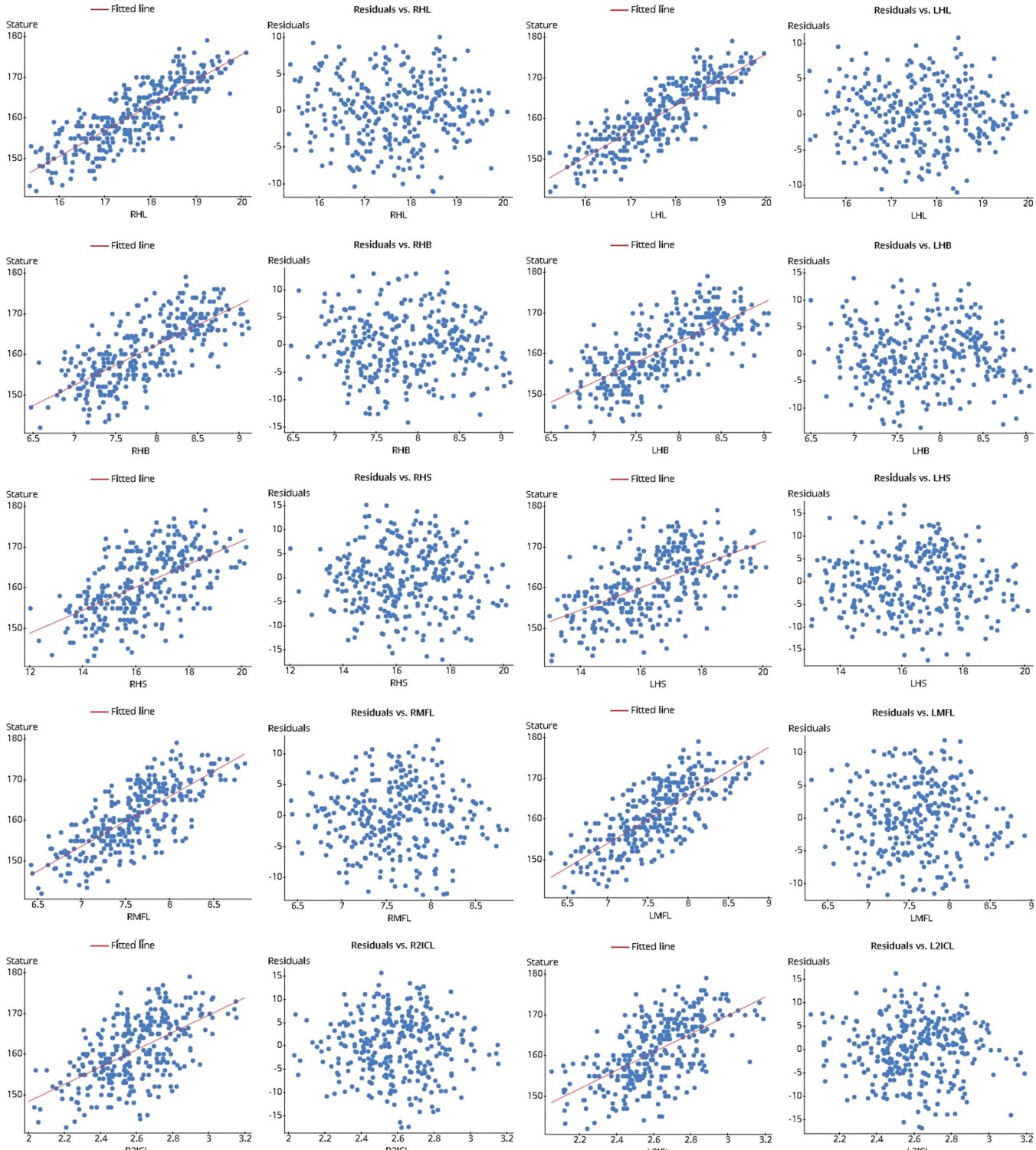

**Fig 4. Scatter plot of the explanatory variables (hand dimensions) with response variables (stature) and residuals against each explanatory variable.**

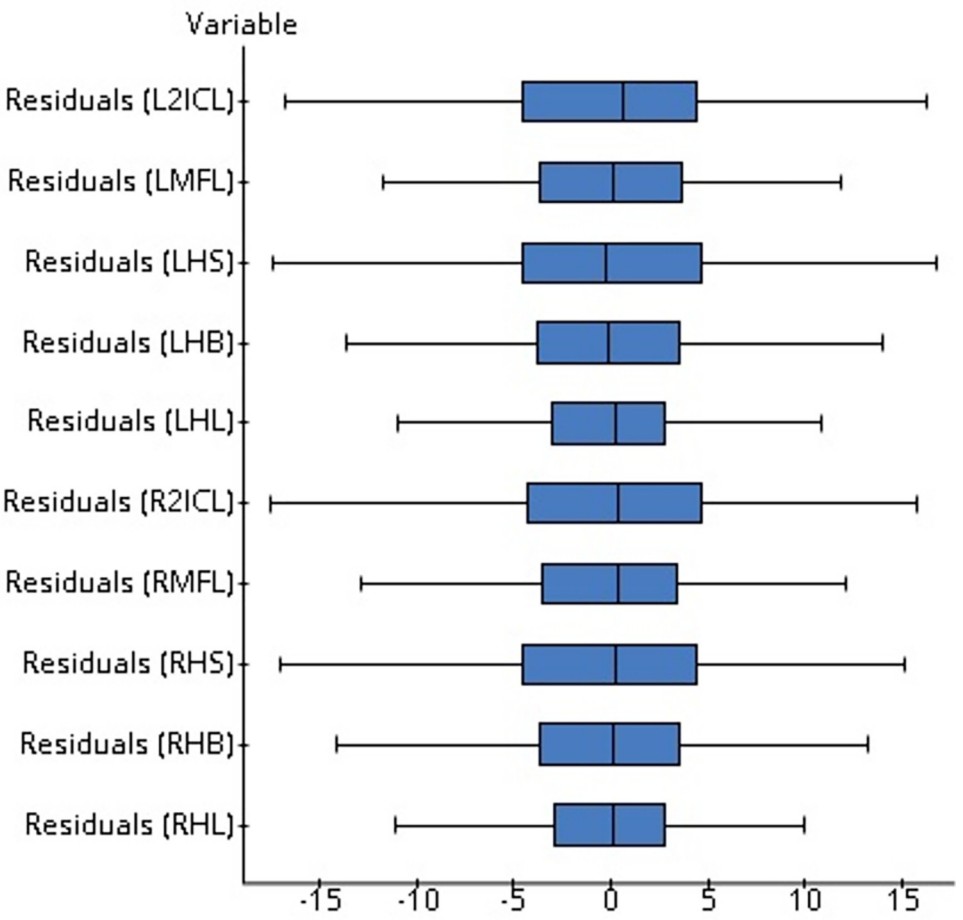

**Fig 5. Boxplot of the residuals against each explanatory variable.**

Bajau Male: Stature = 87.910 + 3.845 LHL+ 2.676 L2ICL
Bajau Female: Stature = 81.960 + 3.845 LHL+ 2.676 L2ICL
Malay Male: Stature = 90.218 + 3.845 LHL+ 2.676 L2ICL

**Table 6. Goodness-of-fit measure.**

| Response variable | Model | $R^2$ | S.E.E. | F |
|---|---|---|---|---|
| **Stature** | 50.402 + RHLx6.260 | 0.6944 | 4.09 | 734.04* |
| | 49.263 + LHLx6.327 | 0.7098 | 3.98 | 789.93* |
| | 83.124 + RHBx9.894 | 0.5282 | 4.11 | 361.62* |
| | 84.494 + LHBx9.786 | 0.5087 | 4.19 | 334.57* |
| | 114.765 + RHS x 2.833 | 0.3227 | 3.74 | 153.87* |
| | 115.298 + LHS x 2.794 | 0.3045 | 3.86 | 141.38* |
| | 67.758 + RMFLx12.267 | 0.5662 | 4.55 | 421.59* |
| | 71.124 + LMFL x 11.829 | 0.5672 | 4.38 | 423.24* |
| | 105.840 + R2ICLx21.252 | 0.3255 | 4.43 | 155.85* |
| | 102.346 + L2ICLx22.503 | 0.3721 | 4.25 | 191.39* |

SEE = Standard error of estimate

* Significance at the $P < 0.001$ level

**Table 7. Correlation between response and explanatory variables (n-368).**

|  | Stature | RHL | LHL | RHB | LHB | RMFL | LMFL | R2ICL | L2ICL | RHS |
|---|---|---|---|---|---|---|---|---|---|---|
| **RHL** | 0.777 | | | | | | | | | |
| **LHL** | 0.785 | 0.988 | | | | | | | | |
| **RHB** | 0.610 | 0.769 | 0.774 | | | | | | | |
| **LHB** | 0.620 | 0.766 | 0.780 | 0.962 | | | | | | |
| **RMFL** | 0.695 | 0.901 | 0.893 | 0.676 | 0.676 | | | | | |
| **LMFL** | 0.707 | 0.897 | 0.907 | 0.692 | 0.693 | 0.962 | | | | |
| **R2ICL** | 0.520 | 0.677 | 0.678 | 0.432 | 0.461 | 0.768 | 0.758 | | | |
| **L2ICL** | 0.539 | 0.700 | 0.710 | 0.460 | 0.478 | 0.765 | 0.775 | 0.902 | | |
| **RHS** | 0.525 | 0.610 | 0.617 | 0.627 | 0.648 | 0.556 | 0.568 | 0.371 | 0.358 | |
| **LHS** | 0.494 | 0.603 | 0.619 | 0.604 | 0.616 | 0.552 | 0.579 | 0.363 | 0.376 | 0.877 |

RHL = Right hand length, LHL = Left hand length, RHB = Right handbreadth, LHB = Left handbreadth, RMFL = Right middle finger length, LMFL = Left middle finger length, R2ICL = Right second inter-crease length, L2ICL = Left second inter-crease length, RHS = Right hand span, LHS = Left hand span

All correlations were significant at p<0.01 level.

Malay Female: Stature = 84.268 + 3.845 LHL+ 2.676 L2ICL
Chinese Male: Stature = 90.218 + 3.845 LHL+ 2.676 L2ICL
Chinese Female: Stature = 84.268 + 3.845 LHL+ 2.676 L2ICL

## Discussion

The present study conducted a multivariate analysis to propose linear models for stature estimation of the young adult population in Sabah. Adopting multivariate analysis attempted to clarify the statistical analysis level. The researchers chose the participants using a stratified random sampling method for four large ethnic groups in Sabah: Malay, Chinese, KadazanDusun, and Bajau. There was an equal number of females and males for each ethnic group. After

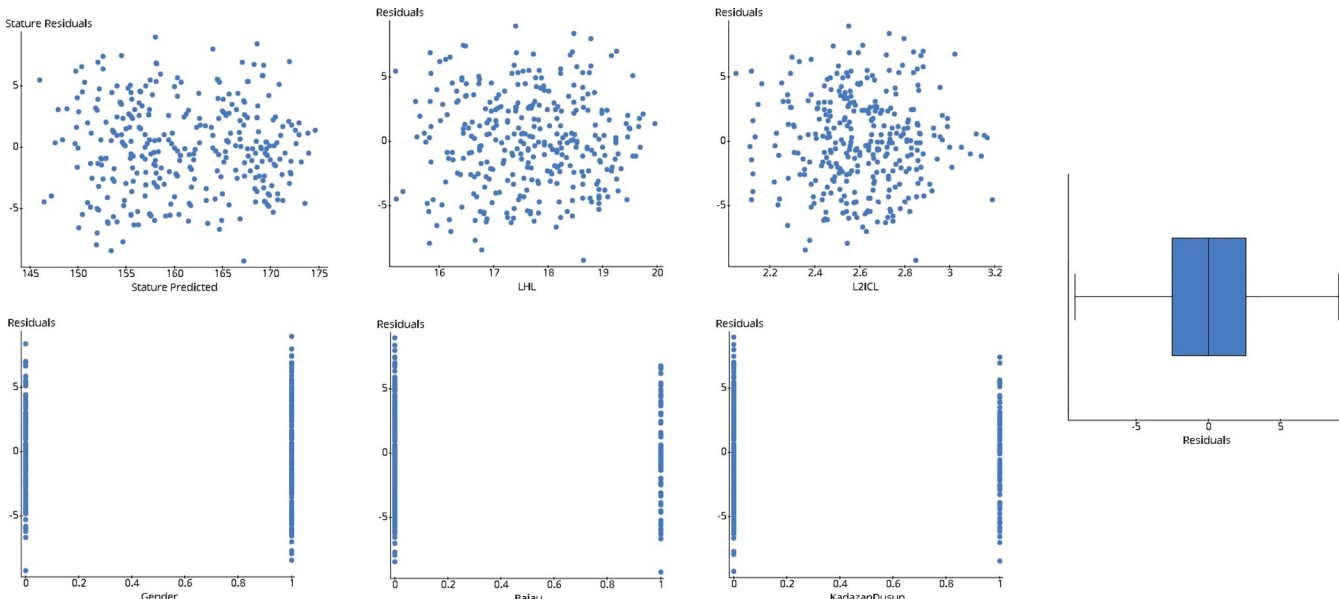

**Fig 6. Scatter plot of stature residuals with predicted stature, sex, Bajau ethnicity, KadazanDusun ethnicity, LHL & R2ICLand boxplot for residuals.**

**Table 8. The forward selection process in multiple regression analysis.**

| Step | Variable | P-value | RMSE | $R^2$ | Adj. $R^2$ |
|------|----------|---------|------|-------|-----------|
| 1 | LHL | <0.0001 | 4.222 | 0.7098 | 0.7089 |
| 2 | Sex | <0.0001 | 3.763 | 0.7702 | 0.7687 |
| 3 | Bajau | 0.0004 | 3.696 | 0.779 | 0.7769 |
| 4 | KadazanDusun | 0.001 | 3.640 | 0.7863 | 0.7837 |
| 5 | L2ICL | 0.062 | 3.626 | 0.7886 | 0.7853 |

randomization, it was ensured that the subjects belonged to a fixed age group with the same nutritional status, i.e., normal BMI. Hence, restriction, randomization, and matching were performed at the study design level to avoid confounding [34].

Almost all the hands' bones finish ossification at 18 years old for both sexes [35]. This study included a fixed age group of 18 to 25 years old. Different studies on university students included 18–25 years to estimate stature from the upper limbs [36–38].

Globally, there is significant variance in height. Western Europe has the tallest countries, whereas Sub-Saharan Africa and Southeast Asia have the shortest [3]. This variation might be due to several factors that influence height, and height is a highly heritable polygenic characteristic. Apart from genetics, various environmental, dietary, hormonal, and socioeconomic factors influence an individual's height, depending on the geographical area and ethnic group [39].

Height is an excellent example of a polygenic hereditary trait altered significantly by environmental influences during fetal life, childhood, and adolescence [40]. Height similarities between relatives show that genetics controls 80% of height variation, with the rest influenced by environmental variables such as food and disease exposure [41]. Environmental factors (including nutrition, sickness, resources, and socioeconomic status) are crucial in determining height, especially during the initial two years of life and in low to middle-income countries [3].

Nutrition is among the essential factors that impact human growth, and poor nutritional intake is linked to growth retardation. The factors affecting negative net nutrition in low-middle income nations are less nutrition, water supply, and sanitation, which leads to malnutrition and the inability of an adult to reach his or her genetic height [42]. A study of different geographical locations on stature among young men in European countries found that diet, notably the intake of high-quality protein, explains most of the differences in height [42]. The present study included the students residing on campus, so the students generally consume the same type of food.

This work investigates the relationship between stature, ethnicity, sex, and hand dimension to examine the potential of using regression equations for predicting the stature of the explanatory variables. This study demonstrates that males have more significant stature and hand dimensions (p<0.001) than females. The results are in line with several works in the literature

**Table 9. Estimates and intercepts of the multiple linear regression model.**

| Parameter | Estimate | Std. Err. | DF | T-Stat | P-value |
|-----------|----------|-----------|-----|--------|---------|
| Intercept | 91.037 | 5.1715143 | 318 | 17.603641 | <0.0001 |
| LHL | 3.777 | 0.37644735 | 318 | 10.032945 | <0.0001 |
| Sex | -5.982 | 0.58144257 | 318 | -10.288616 | <0.0001 |
| Bajau | -2.752 | 0.57774816 | 318 | -4.7628927 | <0.0001 |
| KadazanDusun | -2.118 | 0.56698831 | 318 | -3.7361531 | 0.0002 |
| L2ICL | 2.999 | 1.4405092 | 318 | 2.0817405 | 0.0382 |

**Table 10. Goodness-of-fit measure for multiple linear regression model.**

| Response Variable | Model | $R^2$ | Adj. $R^2$ | F | p-value |
|---|---|---|---|---|---|
| Stature | 90.218 + 3.845 LHL -5.950 Sex—2.308 Bajau -1.673 KadazanDusun + 2.676 L2ICL | 0.7886 | 0.7853 | 238.063 | <0.0001 |

[16, 17, 43], which also mention that sex-sensitive equations must be employed when measuring stature based on different body parts. This study also shows differences in stature measurement for several large ethnic groups. For males, a difference was found between Bajau and Malay, as well as between Bajau and Chinese.

Meanwhile, for females, differences were found between KadazanDusun and Chinese, as well as between Bajau and Chinese. In other studies on West Malaysia, the difference was significant compared to males (p<0.05). For females, a significant difference was seen in stature when compared among Malay and Indian ethnicities (p<0.05) [44].

Previous researchers used simple linear regression to predict stature from different hand dimensions. Additionally, the hand length and handbreadth were used to estimate the stature of Hans' population, with SEE being in the range of 2.95 and 5.64 cm for males and 4.52 to 5.15 cm for females [45]. Other related studies employed handbreadth, max handbreadth, and length of thumb, palm, middle finger, index finger, ring finger, and little finger, with SEE in the range of 5.34 to 6.11 cm for males and 3.68 to 4.39 cm for females [1]. This work employed hand length, handbreadth, middle finger, second inter-crease of the middle finger length, and hand span, with SEE in the range of 3.74 to 4.55 cm.

The present study demonstrated the highest association among explanatory variables for hand length and stature (right side: r = 0.833; left side: r = 0.842). A previous study on 18–25 years old Kashmiri medical students found that the length of the hand was the optimal parameter for stature measurement [46]. Related studies in Western Australia and the Saudi population demonstrated a significant correlation between stature on both sides [44, 45].

The forward selection approach was derived for multiple regression in this work, where a variable was chosen as input for the model in the case of p ≤0.05. The stature estimation models were proposed. Under these criteria, sex, LHL, ethnicity (i.e., Bajau and KadazanDusun), and L2ICL were input to measure the stature. The model explained 78.53% of the variability.

In Sarawak, a study was carried out on the Iban population measurement model to measure stature based on hands and handprints [10]. They did not show the percentage of variability explained by the models. A different work was done on the female Indian population with hand and palm length, and hand and a maximum handbreadth were considered explanatory variables. The model explained 49.1% of the variability of stature [47]. In another study on the Korean population, instead of using hand length, the study used other explanatory variables; the circumference of the wrist, palm, middle finger, and middle finger proximal phalange length of both sexes obtained an $R^2$ value and explained 64.2% of the variability [48]. The $R^2$ value was lower than that in this study. A study on Turks where stature acted as the dependent variable and hand, and foot length as explanatory variables obtained an $R^2$ value of 0.861 [28]. The value was higher than in this study. The probable reason for having a higher $R^2$ is that the foot length is in the formula, as researchers have demonstrated that lower limb dimensions are better predictors of stature than upper limbs [49].

Other than predicting stature from hand dimensions, these measurements can predict the handgrip strength [50]. The hand normative values are helpful for plastic and reconstructive surgery and for specific product design applications like designing garments, gloves, and artificial hand [51].

In the related literature, other upper limbs, humerus, and ulnar are also suitable predictors of stature [12]. Even though this study only estimated hand dimensions to obtain height, it could explain over 60% variability of the Sabahan population. The remaining variability may likely be because of the absence of certain explanatory variables such as age group, ulnar length, radial length, and possibly others. This study was meant to be carried out at the community level. However, due to the Covid-19 pandemic, data gathering at the community level was not possible. Alternatively, this study considered university students who satisfied the inclusion criteria and were vaccinated, who may not represent the major ethnic groups in Sabah. Nevertheless, the participants were from various areas and large ethnic groups in Sabah and had similar lifestyles, eating habits, and traditions. The formulae developed in this study are sex dependent which implies that inability to determine the biological sex would affect the predictability of the formulae. While choosing samples, the medical conditions that may influence anthropometry of the hand and stature were checked using a predefined questionnaire, and no investigation has confirmed this.

## Conclusion

The study proposes linear models for the stature estimation in the Sabahan young adult Population. This study revealed differences, especially between sexes, and some ethnic differences were also found. Hence, the formulae derived for one race and both sexes might not apply to other races and sex. Therefore, this research provides valid anthropometric values for stature measurement based on anthropometric measurement of hand dimension.

## Supporting information

**S1 Data.**
(XLSX)

## Acknowledgments

The authors are grateful to the Student Affairs Department (BPA), UMS for providing the student information. The positive attitude of participants who participated in this project is sincerely appreciated.

## Author Contributions

**Conceptualization:** Sadia Choudhury Shimmi, M. Tanveer Hossain Parash.

**Data curation:** M. Tanveer Hossain Parash.

**Formal analysis:** M. Tanveer Hossain Parash.

**Funding acquisition:** Sadia Choudhury Shimmi, M. Tanveer Hossain Parash.

**Investigation:** Hasanur Bin Khazri.

**Methodology:** Hasanur Bin Khazri, Sadia Choudhury Shimmi, M. Tanveer Hossain Parash.

**Project administration:** Hasanur Bin Khazri, M. Tanveer Hossain Parash.

**Supervision:** Sadia Choudhury Shimmi, M. Tanveer Hossain Parash.

**Writing – original draft:** Hasanur Bin Khazri, M. Tanveer Hossain Parash.

**Writing – review & editing:** Hasanur Bin Khazri, Sadia Choudhury Shimmi, M. Tanveer Hossain Parash.

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
