## [Decision Letter · Decision Letter 0]

8 Jul 2022

PONE-D-22-16753A Multivariate Analysis to Propose Linear Models for the Stature Estimation in the Sabahan Young Adult PopulationPLOS ONE

Dear Dr. Parash,

Thank you for submitting your manuscript to PLOS ONE. After careful consideration, we feel that it has merit but does not fully meet PLOS ONE’s publication criteria as it currently stands. Therefore, we invite you to submit a revised version of the manuscript that addresses the points raised during the review process.

We look forward to receiving your revised manuscript.

Kind regards,

Naji Arafat Mahat, PhD

Academic Editor

PLOS ONE

Journal Requirements:

"The authors are grateful to the Pusat Pengurusan Penyelidikan dan Inovasi (PPI), UMS for the grant to conduct this study and Bahagian Perkhidmatan Akademik (BPA), UMS for providing the student information"

"SCS and MTHP received grant.The Centre for Research and Innovation (PPI), Universiti Malaysia Sabah, funded this research work under the grant "Skim Geran Acculturation" (SGA0041-2019). The funders had no role in study design, data collection and analysis, decision to publish, or preparation of the manuscript."

4. Please include a copy of Table 5 which you refer to in your text on page 11.

Reviewers' comments:

Reviewer's Responses to Questions

**Comments to the Author**

1. Is the manuscript technically sound, and do the data support the conclusions?

Reviewer #1: Partly

Reviewer #2: Yes

Reviewer #3: Yes

2. Has the statistical analysis been performed appropriately and rigorously? 

Reviewer #1: Yes

Reviewer #2: Yes

Reviewer #3: Yes

3. Have the authors made all data underlying the findings in their manuscript fully available?

Reviewer #1: Yes

Reviewer #2: Yes

Reviewer #3: Yes

4. Is the manuscript presented in an intelligible fashion and written in standard English?

Reviewer #1: No

Reviewer #2: Yes

Reviewer #3: Yes

5. Review Comments to the Author

Reviewer #1: I have some major concerns over the scientific basis of the article. Stature is not the definite parameter in biological profiling since it follows normal distribution in each population. It will be more useful if the targeted individual is very tall or very short. Plus the best stature estimator is the femur (lower limb). The article should be written with these facts in mind without 'over-emphasizing' hand anthropometry. Other comments are as outline below.

1. Replace the word "gender" to "sex" throughout the article. Gender should not be used in such studies as gender is a social construct, the correct term is biological sex.

2. Line 71. "Hand morphometry offers crucial evidence in investigating crime scenes"- This statement is not true and needs to be rephrase.

3. Line 81. "The right hand length has precise parameters.."- How precise?

4. Consider diurnal variation - state what time of the day did you took the stature measurements.

4. Line 193. Rephrase the findings of the t-test.

5. Line 204-205. It is acknowledged in the literature that males are taller than females. Rephrase the sentence.

6. Fig 6 should be replaced with a simple correlation table to present the correlation between those five hand dimensions to stature.

7. Ethnicity has to be determined first before application of the formulae developed in this study. Provide demographic data for the people in Sabah.

8. Comparison with hand morphometric data of other ethnic groups in Malaysia. Are they significantly different that warrant specific formulae? Or are they similar that a general formulae for the whole population will be sufficient?

9. Language editing throughout the article, and especially on how to present and discuss the results.

10. Revise the article for grammatical errors, punctuation, and translation to English (Bahagian Perkhidmatan Akademik, Pusat Pengurusan Penyelidikan dan Inovasi).

11. Revise Table 4 and its caption.

12. Line 312-319. Include SD or SEE for each formula.

13. The formulae developed in this study are sex-dependent, what can we do in cases where the sex is unknown?

14. What other areas that may find hand morphometric data valuable? Include in discussion.

15. Line 408-409. Revise the statement.

Reviewer #2: Dear Author,

Thank you for the article. I enjoyed reading it. There are few items which needs to be addressed before the article is publishable. Issues need to be addressed are:

1. Missing stature abbreviations meanings. Example, LHL is not stated anywhere in the text. Not everyone will know left hand length is.

2. Minor grammar issues in Introduction and discussion. Line 74, 76, 81, 332.

3. Table 5 missing from manuscript.

4. Choice of Methodology: Stratified random sampling is acceptable, though I don't think it is suited here. The sample size data collection showcases a fixed amount of participants for each racial group. This is simple random sampling. There is nothing wrong with this methodology as well-when the analysis for each group is conducted in silo. It seems that all analysis are valid except for Table 9. In order to use stratified in this case, you will need to have the correct proportion amount of samples in the group based on the percentage in Sabah's demography to avoid over and under representation of population data.

Good luck.

Reviewer #3: Abstract

line 21- handspan, handbreath, hand?, middle finger length

Introduction

line 68- what kind of personal identification? for individualisation.

line 81- define SEE

limitation and impact of stature due to ethnic variation should be emphasised.

Methodology

why was only the middle fingers were chosen? justify

inclusion criteria - is there any difference between Malay and Malay (Bruneian)?

line 105- it means that the model only applicable for 'pure' ethnic?

Discussion

how did the randomization of sample occur?

line 332, 375 - 18 to 25

line 333 - Western Europe has the tallest countries?

line 399 - COVID-19 or Covid-19, standardize.

line 400 - university

Any example or evidence that show malnutrition or disease cause growth retardation? and how does this correlate with the Sabah ethnic?

Limitation of the proposed formula should be indicated.

6. PLOS authors have the option to publish the peer review history of their article (what does this mean?). If published, this will include your full peer review and any attached files.

Reviewer #1: No

Reviewer #2: No

Reviewer #3: No

---

## [Author Response · Author response to Decision Letter 0]

22 Jul 2022

Response: The revised manuscript meets PLOS ONE’s requirements. 

"The authors are grateful to the Pusat Pengurusan Penyelidikan dan Inovasi (PPI), UMS for the grant to conduct this study and Bahagian Perkhidmatan Akademik (BPA), UMS for providing the student information"

"SCS and MTHP received grant.The Centre for Research and Innovation (PPI), Universiti Malaysia Sabah, funded this research work under the grant "Skim Geran Acculturation" (SGA0041-2019). The funders had no role in study design, data collection and analysis, decision to publish, or preparation of the manuscript."

Response: The authors agreed to remove the sentence having funding statement in the acknowledgement section. The authors do not have any amendment in the current Funding Statement.

Response: The authors have included the Ethics statement in the methodology section.

4. Please include a copy of Table 5 which you refer to in your text on page 11.

Response: The authors have included Table 5 near the table cited.

Review Comments to the Author

Reviewer #1: I have some major concerns over the scientific basis of the article. Stature is not the definite parameter in biological profiling since it follows normal distribution in each population. It will be more useful if the targeted individual is very tall or very short. 

Response: The authors also agree that stature is not the definite parameter in biological profiling. That is why the opening remark is “Stature is one of the significant parameters to confirm a biological profile besides sex, age, and ancestry (Asadujjaman et al., 2019)” 

Plus the best stature estimator is the femur (lower limb). The article should be written with these facts in mind without 'over-emphasizing' hand anthropometry. 

Response: The authors also agree that lower limb parameters are better compared to the upper limb parameters to estimate stature. The authors have included this notion in the discussion. (414-416)

Other comments are as outline below.

1. Replace the word "gender" to "sex" throughout the article. Gender should not be used in such studies as gender is a social construct, the correct term is biological sex.

Response: The authors have amended accordingly.

2. Line 71. "Hand morphometry offers crucial evidence in investigating crime scenes"- This statement is not true and needs to be rephrase.

Response: The authors have omitted the statement

3. Line 81. "The right hand length has precise parameters.."- How precise?

Response: The authors have revised the above quoted part as “right hand length is the most reliable for estimating the stature among these hand measurements”. (Line 82-83)

4. Consider diurnal variation - state what time of the day did you took the stature measurements.

Response: The authors added the statement in the methodology regarding the diurnal variation as “The measurement was taken in the fixed period of 10 am to 12 pm to avoid the possible diurnal variation.” (Line139-140)

4. Line 193. Rephrase the findings of the t-test.

Response: Rephrased as “An independent sample t-test was performed to investigate the statistical significance of the observed difference.” (Line 203)

5. Line 204-205. It is acknowledged in the literature that males are taller than females. Rephrase the sentence.

Response: amended as “Therefore, the result demonstrates that male participants had a higher stature than females.” (Line 215)

6. Fig 6 should be replaced with a simple correlation table to present the correlation between those five hand dimensions to stature.

Response: Authors have replaced Fig 6 with Table 7 (Line 299-303) and subsequent figure number and table number has been amended.

7. Ethnicity has to be determined first before application of the formulae developed in this study. Provide demographic data for the people in Sabah.

Response: The authors have provided demographic data for the people of Sabah (Line:97-105)

8. Comparison with hand morphometric data of other ethnic groups in Malaysia. Are they significantly different that warrant specific formulae? Or are they similar that a general formulae for the whole population will be sufficient?

Response: The hand morphometric data of the participants of the current study are different from other ethnic groups of Malaysia but there was difference in between the major ethnic groups of Sabah. The reasons have been elaborated in the introduction (Line:97-105).

9. Language editing throughout the article, and especially on how to present and discuss the results.

Response: The authors have taken professional service in this regard.

10. Revise the article for grammatical errors, punctuation, and translation to English (Bahagian Perkhidmatan Akademik, Pusat Pengurusan Penyelidikan dan Inovasi).

Response: The authors have amended accordingly. (Line 445)

11. Revise Table 4 and its caption.

Response: The authors have amended accordingly. (Line 248)

12. Line 312-319. Include SD or SEE for each formula.

Response: The authors have mentioned the root mean square error (or RMSE for a regression model is similar with the SD for the ideal measurement model) in Table 8.

13. The formulae developed in this study are sex-dependent, what can we do in cases where the sex is unknown?

Response: The authors have mentioned this issue in the limitation as “The formulae developed in this study are sex dependent which implies that inability to determine the biological sex would affect the predictability of the formulae.” (Lines-430-432)

14. What other areas that may find hand morphometric data valuable? Include in discussion.

Response: The authors have included in the discussion (Line 417-420)

15. Line 408-409. Revise the statement.

Response: The authors revised the statement as “The study proposes linear models for the stature estimation in the Sabahan young adult Population.” (Line 437-438)

Reviewer #2: Dear Author,

Thank you for the article. I enjoyed reading it. There are few items which needs to be addressed before the article is publishable. Issues need to be addressed are:

1. Missing stature abbreviations meanings. Example, LHL is not stated anywhere in the text. Not everyone will know left hand length is.

Response: The authors have tried their best to find all the missing abbreviations and provided accordingly.

2. Minor grammar issues in Introduction and discussion. Line 74, 76, 81, 332.

Response: Authors have amended accordingly.

3. Table 5 missing from manuscript.

Response: The authors have inserted Table 5 in the appropriate place.

4. Choice of Methodology: Stratified random sampling is acceptable, though I don't think it is suited here. The sample size data collection showcases a fixed amount of participants for each racial group. This is simple random sampling. There is nothing wrong with this methodology as well-when the analysis for each group is conducted in silo. It seems that all analysis are valid except for Table 9. In order to use stratified in this case, you will need to have the correct proportion amount of samples in the group based on the percentage in Sabah's demography to avoid over and under representation of population data.

Response: As the subjects were first stratified by ethnicities and then by gender, and then they were randomly selected, this is a stratified random sampling. In the assumption for conducting multiple linear regression, the subjects should be selected through any of the random sampling techniques (Sullivan III, 2017) and each stratum should have the minimum sample size of 25 (Jenkins and Quintana-Ascencio 2020). As this study included samples (46 per stratum) more than the minimum sample (25 per stratum) required for multiple linear regression and having same number participants facilitates homogeneity of the stratum which is more desirable than sample being proportion to the population. As the subjects were included in the study through stratified random sampling, every person within the sampling frame had the equal probability to be included in the study which addressed the representation of the population. 

Good luck.

Reviewer #3: Abstract

line 21- handspan, handbreath, hand?, middle finger length

Response: The authors have inserted the missing word “length”. (Line 20)

Introduction

line 68- what kind of personal identification? for individualisation.

Response: When DNA fingerprinting requires a large data to compare with the crime scene findings. By estimating stature, it would narrow down the focus point which would aid to identify the victim. (Krishan et al.,2012)

line 81- define SEE

Response: Authors have included the definition. (Line 87)

limitation and impact of stature due to ethnic variation should be emphasised.

Response: The authors have followed the suggestion and amended accordingly (Line 96-106)

Methodology

why was only the middle fingers were chosen? Justify

Response: The anatomical axis of the hand passes through the middle finger. Other than that compared to other fingers the middle finger demonstrated higher relationship with the stature. That is why the researchers chose the middle finger.

inclusion criteria - is there any difference between Malay and Malay (Bruneian)?

Response: The authors have mentioned the difference in the introduction (Line 100-102)

line 105- it means that the model only applicable for 'pure' ethnic?

Response: Yes. The researchers included the participants whose parents and grandparents belonged to the same ethnic group. 

Discussion

how did the randomization of sample occur?

line 332, 375 - 18 to 25

Response: Through stratified random sampling. The subjects were first stratified by ethnicities and then by gender, and then they were randomly selected.

line 333 - Western Europe has the tallest countries?

Response: The literature review shows that Western Europeans are comparatively taller than other part of the world.

line 399 - COVID-19 or Covid-19, standardize.

Response: The authors have standardized to Covid-19.

line 400 – university

Response: The authors have amended accordingly

Any example or evidence that show malnutrition or disease cause growth retardation? and how does this correlate with the Sabah ethnic?

Response: Kyle UG, Shekerdemian LS, Coss-Bu JA. Growth failure and nutrition considerations in chronic childhood wasting diseases. Nutr Clin Pract. 2015 Apr;30(2):227-38. doi: 10.1177/0884533614555234. Epub 2014 Nov 6. PMID: 25378356. This article mentioned about malnutrition and various diseases that cause growth retardation. 

The conditions and disease that are related to growth retardation are applicable to all population and ethnicities. This is how this correlated with Sabah ethnic groups.

Limitation of the proposed formula should be indicated.

Response: Authors have indicated the limitations of the proposed formula. (Line 431-433)

---

## [Decision Letter · Decision Letter 1]

15 Aug 2022

PONE-D-22-16753R1A Multivariate Analysis to Propose Linear Models for the Stature Estimation in the Sabahan Young Adult PopulationPLOS ONE

Dear Dr. Parash,

Thank you for submitting your manuscript to PLOS ONE. After careful consideration, we feel that it has merit but does not fully meet PLOS ONE’s publication criteria as it currently stands. Therefore, we invite you to submit a revised version of the manuscript that addresses the points raised during the review process.

We look forward to receiving your revised manuscript.

Kind regards,

Naji Arafat Mahat, PhD

Academic Editor

PLOS ONE

Journal Requirements:

Reviewers' comments:

Reviewer's Responses to Questions

**Comments to the Author**

1. If the authors have adequately addressed your comments raised in a previous round of review and you feel that this manuscript is now acceptable for publication, you may indicate that here to bypass the “Comments to the Author” section, enter your conflict of interest statement in the “Confidential to Editor” section, and submit your "Accept" recommendation.

Reviewer #1: (No Response)

Reviewer #2: All comments have been addressed

Reviewer #3: All comments have been addressed

2. Is the manuscript technically sound, and do the data support the conclusions?

Reviewer #1: Yes

Reviewer #2: Yes

Reviewer #3: Yes

3. Has the statistical analysis been performed appropriately and rigorously? 

Reviewer #1: Yes

Reviewer #2: Yes

Reviewer #3: Yes

4. Have the authors made all data underlying the findings in their manuscript fully available?

Reviewer #1: Yes

Reviewer #2: Yes

Reviewer #3: Yes

5. Is the manuscript presented in an intelligible fashion and written in standard English?

Reviewer #1: No

Reviewer #2: Yes

Reviewer #3: Yes

6. Review Comments to the Author

Reviewer #1: Dear authors,

Content-wise, I am happy with the revised manuscript.

However, the manuscript needs to undergo another round of language editing. Revise the usage of past tense and past perfect tense in the manuscript. Grammatical errors need to be corrected:

1. Replace "gender" with "sex".

2. Replace "scoping down" with "narrowing down".

3. Line 86: There is no Reference no. 132.

4. Line 123

5. Line 136

6. Line 139

7. Line 156

8. Line 168

9. Line 253-254

10. Line 352

11. Line 355

12. Line 439

and throughout the manuscript.

Reviewer #2: The author has addressed the issues raised by the current reviewer previously and the paper is now acceptable for publication.

Reviewer #3: All comments have been addressed. It is a pleasure to read the manuscript. It is a very interesting field to enhance the forensic investigation. Thank you.

7. PLOS authors have the option to publish the peer review history of their article (what does this mean?). If published, this will include your full peer review and any attached files.

Reviewer #1: No

Reviewer #2: No

Reviewer #3: No

---

## [Author Response · Author response to Decision Letter 1]

15 Aug 2022

Dear editor and reviewers,

Please accept our sincere gratitude for your volunteerism to review our manuscript to polish it to be fit for the renowned journal. We do appreciate your valuable time and critical thinking for the purpose to enrich the manuscript. Each of your comments are very crucial and we have tried our best either to incorporate into the manuscript or to answer to your queries. 

The following are the responses prepared by the authors and have been inserted issue by issue: 

Review Comments to the Author

Reviewer #1: Dear authors,

Content-wise, I am happy with the revised manuscript.

However, the manuscript needs to undergo another round of language editing. Revise the usage of past tense and past perfect tense in the manuscript. Grammatical errors need to be corrected:

1. Replace "gender" with "sex".

Response: All "gender" have been replaced with "sex"

2. Replace "scoping down" with "narrowing down".

Response: Amended as per advice

3. Line 86: There is no Reference no. 132.

Response: It is actually 13. There was a typo when the references were re-numbered.

4. Line 123

5. Line 136

6. Line 139

7. Line 156

8. Line 168

9. Line 253-254

10. Line 352

11. Line 355

12. Line 439

and throughout the manuscript.

Response: The authors have taken the assistance of grammar correcting software Grammarly and also taken the editing service from PM Proofreading Services. The certificate provided by them has been attached.

Reviewer #2: The author has addressed the issues raised by the current reviewer previously and the paper is now acceptable for publication.

Reviewer #3: All comments have been addressed. It is a pleasure to read the manuscript. It is a very interesting field to enhance the forensic investigation. 

Our sincere gratitude to all the reviewers for your sincere contribution for the betterment of science. At the end, we hope that our sincere effort could address all the requirements of the editor and our revision would satisfy the imminent reviewers. 

Sincerely yours,

M Tanveer Hossain Parash (corresponding author)

---

## [Editor Report · Decision Letter 2]

17 Aug 2022

A Multivariate Analysis to Propose Linear Models for the Stature Estimation in the Sabahan Young Adult Population

PONE-D-22-16753R2

Dear Dr. Parash,

We’re pleased to inform you that your manuscript has been judged scientifically suitable for publication and will be formally accepted for publication once it meets all outstanding technical requirements.

Kind regards,

Naji Arafat Mahat, PhD

Academic Editor

PLOS ONE
---

## [Editor Report · Acceptance letter]

19 Aug 2022

PONE-D-22-16753R2 

 A Multivariate Analysis to Propose Linear Models for the Stature Estimation in the Sabahan Young Adult Population 

Dear Dr. Parash:

I'm pleased to inform you that your manuscript has been deemed suitable for publication in PLOS ONE. Congratulations! Your manuscript is now with our production department. 

Kind regards, 

on behalf of

Dr. Naji Arafat Mahat 

Academic Editor

PLOS ONE